# Folliculogenesis: A Cellular Crosstalk Mechanism

**DOI:** 10.3390/cimb47020113

**Published:** 2025-02-10

**Authors:** Bianca Viviana Orozco-Galindo, Blanca Sánchez-Ramírez, Cynthia Lizeth González-Trevizo, Beatriz Castro-Valenzuela, Luis Varela-Rodríguez, M. Eduviges Burrola-Barraza

**Affiliations:** 1Facultad de Zootecnia y Ecología, Universidad Autónoma de Chihuahua (UACH), Perif. Fco. R. Almada Km. 1, 31453 Chihuahua, Mexico; p384262@uach.mx (B.V.O.-G.); bcastro@uach.mx (B.C.-V.); 2Facultad de Ciencias Químicas, Universidad Autónoma de Chihuahua (UACH), Campus Universitario #2, 31125 Chihuahua, Mexico; bsanche@uach.mx (B.S.-R.); lvrodriguez@uach.mx (L.V.-R.); 3School of Engineering and Sciences, Tecnologico de Monterrey, Av. H. Colegio Militar 4700, Nombre de Dios, 31300 Chihuahua, Mexico; cynthial.gonzalez@tec.mx

**Keywords:** folliculogenesis, crosstalk, oocyte, follicle, embryo

## Abstract

In vitro embryo production has accelerated in the cattle industry in recent years. Because women are similar to cows, this represents an opportunity to improve women’s reproductive protocols. This review focuses on crosstalk communication during folliculogenesis for an in-depth understanding of the events involved in developing the oocyte competence necessary to generate an embryo after fertilization. This knowledge can be used to improve oocytes in in vitro maturation cultures, which would allow us to obtain oocytes of high quality and competence, resulting in successful pregnancies in both women and cows. The first part of this review covers the concepts of cellular crosstalk before puberty in the primordial, primary, and secondary follicles. The next part involves cellular crosstalk after puberty, when gonadotropin hormones act on the ovary, promoting oocyte maturation. The final part comprises a perspective on using cow models to study human ovary physiology.

## 1. Introduction

According to the Data Retrieval Committee of the International Embryo Technology Society, IETS, https://www.iets.org/, (accessed on 20 August 2024), in vitro technology is among the most essential cattle embryo production techniques. The statistics show that in the last fourteen years, global in vitro embryo cattle production has increased by 250%, displacing in vivo embryo-derived production, which has declined by 43% (Figure 1). These data reveal the importance of the cattle embryo industry, even more so in a world where, according to the Organization of the United Nations (ONU), about 800 million people suffer from hunger. Thus, in vitro technology offers the possibility of generating animals to contribute to the nutrition of people around the world. The bovine model presents an opportunity to study the whole range of in vitro embryos, from oocyte maturation through fertilization to embryo stages such as zygotes, morulas, and blastocysts.

On the other hand, early embryonic development is remarkably preserved in mammals. Hence, an inherent benefit of the bovine animal model is the possibility of extrapolating all the knowledge harvested in embryo production to humans. It is worth mentioning that in humans, the study of assisted reproduction is complex because of ethical questions and, in addition, the obtention of cellular material is complicated, which results in the low competence of embryo development with low percentages (30%) of live births per cycle [1,2].

The mammalian ovary is extraordinary machinery in which the cycle of ovarian activity happens. This small organ, with a volume of approximately six cm^3^ in both cows and women, plays a crucial role in the reproductive system of mammals, undergoing regular changes in response to hormonal fluctuations [3]. This machinery is a highly ordered process influenced by a milieu of factors from various physiological stimuli, such as interns between cells and externs by the hypothalamic–pituitary–gonadal axis [4]. Understanding the intricacies of ovarian function is essential to comprehending reproductive processes in mammals. During the folliculogenesis stage, the follicles grow, and the cells inside also change. The follicle compartment contains several types of cells, each with a different metabolism and function [5]. After embryonic development, the follicle grows from a primordial stage until the antral stage in puberty. At the same time, the oocyte gradually increases in size while the surrounding follicular cells grow exponentially from an initial monolayer of flattened pre-granulosa cells to a multilayer of stratified mural and cumulus cells [6]. In the past few decades, several studies have provided insight into how the oocyte establishes and maintains contact with the cells of the ovarian follicle that enclose it [7,8,9]. Communication between the cells comprising the follicle consortium is vital during folliculogenesis. It is present as cellular crosstalk in all the follicle stages in a coordinated and specific manner to conduce to the competence of the oocyte [10].

Taking knowledge obtained from mouse ovaries as a basis and complementing this with knowledge from humans and cows, this review focuses on cellular crosstalk communication during the folliculogenesis process for an in-depth understanding of the events involved in the success of oocyte competence that is necessary to generate an embryo after fertilization.

## 2. Folliculogenesis

In the mammalian ovarian follicular compartment, different cell types interact with each other in ways that are so interdependent that the follicle has been compared to a syncytium. In this structure, there is an ease of communication between the cell types that compose it [10]. Oogenesis begins during fetal life, when primary oocytes, arrested in the diplotene from prophase I of meiosis I, are produced from the germ cell nest and continue after birth until the end of the female reproductive period [11]. This results in the growth and differentiation of the ovarian follicles from the primordial follicle stage through the primary, secondary, and Graaf follicle stages [12].

### **2.1.** Cellular Crosstalk in the Beginning

When the primary oocytes are formed, they are surrounded by squamous pregranulosa cells to form the primordial follicle. In mammal species such as humans and cows, the primordial follicle has a diameter of around 40 μm, with an oocyte of 30 μm surrounded by about 10 pregranulosa cells [13,14]. These cells surround the oocyte, which begins enlarging and conforms to the primordial follicle, which is separated from the rest of the ovary by the basal lamina and remains dormant [4,10]. During this time, there is a great loss of primary oocytes, in which approximately less than 20% (humans) and 5% (cows) of these cells survive to conform to the primordial follicle [15]. These primordial follicles remain in a dormant state because the phosphatase and tensin homolog delete on chromosome 10 (PTEN) revert phosphatidylinositol 3,4,5-trisphosphate (PIP3) to phosphatidylinositol 4,5-biphosphate (PIP2), suppressing PI3K signaling, which maintains the repressor forkhead transcription factor FOXO3 in the nucleus, and blocking genetic expression [16]. The total number of primordial follicles constitutes the finite ovary reserve, which means they are not replaceable [17]. At birth, the number of primordial follicles differs among mammal species; for example, in cows, there are around 135,000, and in humans, there are 700,000 [18]. This ovarian reserve will decrease throughout one’s life until the ovary is left without oocytes. Eventually, 99% die by atresia as females progress through their reproductive lives.

In this stage of folliculogenesis, a phenotypic change occurs in pregranulosa cells. They become proliferative, forming a single layer with approximately 40 cuboidal granulose cells (GCs) for cows [14] and 100 for humans [19]. A quiescent primordial follicle is activated to grow, conforming to the primary follicle stage. This primordial to primary follicle transition has also been called the initial recruitment. This activation has been intensely studied in mouse models, in which it is controlled by the action of the target of the rapamycin complex 1 (mTORC1) signaling pathway in pregranulosa cells. This secretes the protein KITL, which actuates over the Kit receptor on the oocyte membrane and triggers phosphatidylinositol 3 kinase (PI3K) to provoke the conversion of PIP2 to PIP3. This in turn leads to the recruitment of phosphatidylinositol-dependent kinase 1 (PDK1) to activate protein kinase B (AKT) signaling, which promotes the shuttle of the repressor FOXO3 from the nucleus into the cytoplasm of the oocyte, which allows for gene expression for oocyte growth [16]. In humans, it is established that a quickening rate of primordial follicle activation is a cause of premature ovarian insufficiency (POI), which can result in a diminution of the ovarian reserve [20]. In this regard, in vitro studies using ovarian tissue cultures, in both humans and cows, have demonstrated that using bbV (HOpic) as an inhibitor of PTEN increased the activation of primordial follicles; however, in humans, the follicles’ survival rates were severely affected [21,22]. Hence, further in-depth studies are required to clarify primordial follicle activation in humans and cows.

In the primary follicle stage, the oocyte begins to secrete paracrine factors such as bone morphogenetic protein 15 (BMP15) and growth differentiation factor 9 (GDF9), which act on the granulosa cells, triggering the SMAD pathway, which turns on the expression of the genes involved in the proliferation, glycolysis, and cholesterol pathways [23,24]. Also, BMP15 and GDF9 stimulate the gene expression of the follicle-stimulating hormone receptor (FSHR) [25]. Both GDF9 and BMP15 bind with the bone morphogenetic protein type II (BMPR2) receptor. However, for GDF9, BMPR2 forms a complex with activin receptor-like kinase 5 (ALK5), triggering the SMAD2/3 pathway, and for BMP15, the complex is BMPR2, with activin receptor-like kinase 6 (ALK6) activating SMAD1/5/8 signaling [26]. GDF9 and BMP15 can also heterodimerize to form the growth factor cumulin, which potentiates SMAD2/3 signaling [27] (Figure 2). In bovine cumulus cells, the functionality of the BMR2 receptor is regulated by the action of miR-187, which binds to the 3′UTR of *Bmpr2* mRNA and blocks its translation. Moreover, the action of miR-187 is controlled by the circular RNA ciRS-187, which acts as a molecular sponge for miR-187, promoting the overexpression of the BMPR2 receptor [28].

At the same time, the oocyte has elevated mRNA synthesis, and cortical granules are organized. Likewise, it initiates the production of the zona pellucid (ZP), which is composed of three glycoproteins named ZP1, ZP2, and ZP3, which conform to a matrix, like a thick extracellular coat surrounding the oocyte membrane that physically separates the oocyte from the adjacent GCs and functions as a barricade during sperm fertilization [29] (Figure 2). Also, the GCs synthesize growth factors such as epidermal growth factor (EGF), fibroblast growth factor (FGF), and anti-Müllerian hormone (AMH). At this point, AMH is of special importance in regulating folliculogenesis. This hormone triggers its effect across the anti-Müllerian hormone receptor type II receptor (AMHR2) in conjunction with any of the ALK2/ACVR1, ALK3/BMPR1A, or ALK6/BMPR1B receptors to activate by phosphorylation the SMAD1, SMAD5, and SMAD8 proteins, which interact with SMAD4, leading to a complex that translocates into the nucleus to act as a transcription factor over target genes [30]. In women and cows, the AMH levels in serum are used as markers of the growing pool of follicles; in other words, the presence of AMH in serum is correlated with a healthy ovarian reserve [30,31]. The opposite is observed in women with POI, whose AMH levels are low or practically undetected [32]. Simultaneously, the Hedgehog (Hh) pathway starts to direct the specification of thecal cells (TCs), which form a basement membrane around the outermost GC layer, and cells from the surrounding ovarian stroma are condensed around this to create a layer. The TC layer becomes increasingly vascularized, while the GC layer remains avascular [4]. Thus, the secondary follicle is formed (Figure 2), conforming to an oocyte surrounded by layers of granulosa cells. Both humans and bovines show secondary follicles with sizes near a diameter of 120 μm, with an oocyte around 50 μm in diameter enclosed by three to five layers of granulosa cells [14,33,34]. In cows, these layers comprise 41 to 100 granulosa cells [14].

As the secondary follicle is enlarged, the GCs produce and secrete TGF-β-ligands such as activin and inhibin, and each one acts as a dimer like a peptide or hormone in autocrine–paracrine form, regulating follicular development. There are two mature activins, activin-A composed of β_A_β_A_ subunits, and activin-B, composed of β_B_β_B_ subunits. Inhibins also have two mature variants: inhibin-A, which is constituted by αβA, and inhibin-B, which is constituted by αβB. Activins execute their actions by activin type II receptors (ActRIIA or ActRIIB) and activin type I receptors (ActRIB or ALK4) to trigger the SMAD2/3 pathway, which culminates with the activation of genes such as *Esr1*, *Esr2*, *Kitl*, *Taf4b*, and *c-Myc*. Inhibin blocks the activins’ action via a competitive bind to the ActRIIB receptor [35]. The GCs start proliferating under the stimuli of activin in conjunction with the NOTCH pathway [17] (Figure 3). Activin A action was evaluated in an in vitro culture of fetal bovine ovarian cortical pieces, which increased the formation of primordial and primary follicles. Also, in fetal bovine ovaries, the presence of activin A was detected by immunohistochemistry in oocytes and granulosa cells of primordial, primary, and secondary follicles. These results indicate that activin A is active from the early follicle stages, forming and activating them [36].

Consequently, in the secondary follicle, transzonal projections (TZPs) are formed from GCs, which are specialized filopodia structures of about one μm, composed of actin or tubulin filaments that cross the ZP and touch the microvilli on the surface oocyte membrane, establishing communication and maintaining the adhesion between these cells [37]. In GCs from mice and humans, the formation of TZPs is established by the action of the protein myosin-X (MYO10), which can bind both to actin via its head motor domain and to tubulin by its MyTH4-FERM domain [38,39] (Figure 3). Moreover, double-membrane-spanning hydrophilic channels named gap junctions establish direct cell-to-cell communication because they connect the cytoplasmic compartment between follicular cells and the oocyte. Each gap junction is formed by two hemichannels forming a docking called connexons, in which each one is on a cell plasma membrane that is face-to-face with another cell plasma membrane. Each connexon has a hexameric construction with a toroid appearance composed of six proteins termed connexins organized around an aqueous pore [40]. Connexins 43 (Cx43) form gap junctions between GCs, whereas those that connect the oocyte to the surrounding GCs are composed of connexins 37 (Cx37) located at the apex of TZPs. Through these gap junction channels, it is possible to direct the passage of small molecules of approximately one kDa in size, for example, amino acids (glycine, alanine, lysine, and taurine), inositol phosphate, cyclic nucleotides, miRNAs, or secondary messengers such as Ca^2+^, cAMP, and cGMP. Also, even peptides 1.8 kDa in size can be disseminated across gap junctions [41,42]. Other metabolites that can pass through gap junctions are derived from cholesterol and glycolysis. The cholesterol synthesis and glucose degradation stimulus are achieved in GCs when the oocyte secretes BMP15 and GDF9 and binds on the GC’s membrane with BMPRII/ALK (Figure 3). The oocyte cannot synthesize cholesterol de novo because it is deficient in the transcripts encoding the enzymes for this metabolite and lacks a high-density lipoprotein (HDL) receptor, so it is unable to take the cholesterol from the extracellular microenvironment; the only way to obtain it is from GCs through gap junctions [43]. In the case of glucose, the oocyte can uptake glucose by its glucose transporter (GLUT) proteins but has a low capacity to utilize this metabolite as a substrate due to having a limited amount of the glycolytic enzyme phosphofructokinase. In contrast, GCs contain more GLUT proteins than the oocyte and have a high phosphofructokinase activity, so they quickly convert glucose into substrates such as pyruvate, lactate, and NADPH, which are readily diffused across the gap junctions to the oocyte to supply its deficiencies [42] (Figure 3).

In mammals, the gap junctions also promote oocyte arrest in prophase I of meiosis I. Prophase I is maintained by the cyclic adenosine monophosphate (cAMP) level, which is increased because the phosphodiesterase 3A (PDE3A) activity is inactivated by cyclic guanosine monophosphate (cGMP) action. The cGMP is produced in the GCs and can pass through the gap junction formed by connexins-37 (Cx37) to the oocyte cytoplasm or oolemma. It is essential to mention that the oocyte leads this phenomenon because it secretes GDF9 and BMP15, which act on their receptor in the GCs and trigger the pathway to express the protein natriuretic peptide precursor (NPR2). NPR2 is a membrane protein that acts as a receptor for the c-type natriuretic peptide (CNP) secreted by the GCs. Thus, the CNP acts as an autocrine factor on GCs, inciting a series of signaling to produce the cGMP needed for the oocyte’s arrest [44]. Likewise, the CNP/NPR2 determines the expression of proteins, as MYO10 is related to TZP assembly, which strengthens the crosstalk communication between the GCs and the oocyte [45]. In the oocyte, cAMP triggers the protein kinase A (PKA) pathway, provoking the phosphorylation of the cell division cycle 25B (CDC25B) and the oocyte-specific kinase WEE1B. Once WEE1B is phosphorylated, this kinase provokes the phosphorylation in the CDK1 protein, which inactivates the maturation-promoting factor MPF, composed of CDK1 plus cyclin-B protein. Thus, an arrest occurs in prophase I, which is maintained until puberty [46].

### 2.2. Cellular Crosstalk at the End

A series of environmental changes accompany the arrival of puberty. One of these changes is caused by the hypothalamus in the brain with the release of the gonadotropin-releasing hormone (GnRH). The GnRH hormone acts on the pituitary gland, activating the *Cga* and *Fshβ* genes, whose expression conforms to the follicle-stimulated hormone (FSH) subunits. On the other hand, the GDF9 protein secreted by the oocyte induces the expression of the receptor FSHR in the granulosa cells to match that of the FSH. The activin secreted by GCs migrates to the brain, where it acts on the pituitary gland, overstimulating the expression of the *Fshβ* gene. As soon as the FSH is synthesized, it enters the bloodstream and reaches the ovary, where it finds its receptor, the FSHR, in the membrane of the GCs [47]. This union causes the activation of adenyl cyclase to produce cAMP, which turns on the PKA pathway and culminates with the activation of transcription factors such as CREB, MEK1/2, P38, JNK, and GATA4/6, which turns on the transcription of genes such as *Fshr, Lhcgr*, *Cyp19a1*, *Cyp11a1*, and *Hsd3b1*. Also, the union of FSH/FSHR acts on the adapter protein APPL1, which turns on PI3K and incites the AKT signaling pathway, subsequently acting on the movement of calcium ions and inactivating the FOXO1A protein. This singularity provokes the upregulation of genes involved in cellular proliferation, apoptosis inhibition, and atresia prevention [48]. Due to the effect of FSH, the GCs can produce estradiol by expressing the aromatase cytochrome P450 family 19 subfamily A member 1 (CYP19A1). Then, estradiol is secreted and acts as an autocrine–paracrine factor on the GCs, promoting protein expression such as that of cyclin D2, leading to follicle growth. Also, estradiol encourages the expression of NNPC and NPR2 in GCs, which maintains the arrest stage in prophase I of the oocyte [49]. The estradiol secreted travels to the hypothalamus and acts on kisspeptin neurons to induce the expression of GnRH. Next, the GnRH increases luteinizing hormone (LH) and decreases FSH secretion [48].

On the other hand, in the secondary follicles, a cavity named the antrum transforms them into tertiary or antral follicles (Figure 4), which cause the GCs to differentiate into mural (MC) and cumulus (CC) cells. The antrum is filled with follicular fluid, which is composed of proteins, metabolites, ions, hormones, lipids, energy substrates, reactive oxygen species, and extracellular vesicles (EVs) that come from the bloodstream of thecal capillaries located in the cortical region of the ovary and from the secretion of GCs (mural and cumulus cells). The follicular fluid components can be distributed throughout the follicular consortium and interact with the follicular cells and the oocyte, activating events such as signaling pathways, gene expression, and cellular remodeling. Thus, the follicular fluid becomes a reserve of molecules needed to culminate in the folliculogenesis process [37]. Of all the components of follicular fluid, EVs stand out because they are phospholipid bilayer vesicles packed with proteins, miRNAs, lncRNAs, mRNAs, and lipids. If the EVs have a diameter range of 100–1000 nm, they are called microvesicles, but if the diameter varies from 30 to 100 nm, they are labeled as exosomes (Figure 4). The EVs serve as vesicle cargo that can transport their contents to a target cell by endocytosis with the interaction of a membrane receptor by a direct fusion with the cell membrane or by extracellular protease actions that degrade their membrane to release their content and bind to the receptor of a target cell [6].

Cryo-transmission electron microscopy (Cyo-TEM) studies in healthy human follicular fluid have surprisingly revealed that EVs have a diverse morphology. They can be single vesicles, oval vesicles, or double vesicles, with a small vesicle inside a large vesicle. Moreover, EVs can exist as triple to sextuple vesicles, which consist of two small vesicles trapped in a large one. Similarly, EVs are present as vesicle sacs, which are, in turn, full of small vesicles with double-membrane bilayers. EVs can also take the form of tubule vesicles, pleomorphic membrane structures, and lamellar bodies [50]. The lipid compositions in follicular fluid EVs of humans and cows are similar, including cholesteryl ester, cholesterol, ceramides, sphingomyelin, phosphatidylinositol, and phosphatidylcholine, to name a few [51,52]. As for proteins, in human follicular fluid, EVs contain around 662–1000 proteins [52,53], while in bovines, they contain about 322 proteins [54]. The miRNA profile analysis has shown that these bovine EVs contain around 280 miRNAs [55], and human EVs enclose about 192 of these molecules [56]. Of all the miRNAs that have been reported in EVs, those that have been associated with the acquisition of oocyte competence are distinguished. For bovines, bta-miR-34c, bta-miR-141, bta-miR-200a/b, and bta-miR-2285aa stand out; for humans, miR-92a, has-miR-214, has-miR-145, and has-miR-454 are highlighted [56,57]. In bovines, an in-depth analysis of EVs with diameters between 30 and 200 nm derived from ovarian antral follicles using RNA-seq and real-time PCR screened for the presence of mRNAs such as *Dntm1*, *Dntm3A*, *Ehmt1*, *Hdac2*, *Eif4b*, and *Eif4e,* which are involved in the DNA methylation process [55]. Additionally, the supplementation of culture media with these EVs during the in vitro maturation of COCs improved the competence quality of oocytes, which showed high methylation levels and a significantly increased blastocyst rate compared with the control culture [55]. Similarly, the supplementation of culture media with bovine follicular fluid EVs (150 nm) not only promoted bovine GCs’ proliferation but also inhibited cell apoptosis, enhanced the secretion of estradiol and the mRNA expression of *Fshr*, *Cyp19a* and *Bcl2,* and decreased the *Bax* mRNA expression [58].

The MCs are found in a thin layer that lines the inside of the follicular wall. The CCs form a thicker cloud surrounding the oocyte and show TZPs across the zona pellucida, reaching the oolemma and conforming to the complex oocyte cumulus (COC) [10]. The COC forms a visibly distinguishable complex, comprising three to four layers of approximately 2000 CCs, tightly packed together in concentric layers enveloping the oocyte [59]. On MCs, the stimulus of FSH, together with the stimulus of TGF-β and estradiol, causes an increase in the CNP levels, which, after being secreted, triggers cGMP production in the CCs; consequently, the cAMP level is increased in the oocyte in such a way that the arrest in prophase I continues [60]. The intraovarian FSH responsiveness contributes to the conformation of cohorts of growing antral follicles, a phenomenon known as follicular waves. In cows and women, an ovarian cycle could have two to three follicular waves, whereas each cohort could have 8 to 41 follicles in bovines and 4 to 14 follicles in humans [61]. Only one will be selected as dominant from all these follicles, the Graffian follicle, and the rest will die by atresia. Atresia is a mechanism that refers to the degeneration and resorption of follicles by a noninflammatory process before ovulation. The most common form of atresia occurs through apoptosis, although it also can be provoked by the degenerative process of necrosis [62]. Follicle selection is related to the acquisition of LH dependence, which is correlated with the presence of the receptor for LH (LHR) on theca and MC cells, which is induced by the FSH action [63].

Once the LH arrives at the ovary (Figure 4), it binds to the LHR on theca cells. The LH/LHR stimulates an increase in the cAMP levels, which activates the PKA pathway, culminating in the expression of genes related to androstenedione production from cholesterol. The androstenedione passes to the MCs and is processed by CYP19A1, generating more estradiol. On the MCs, the LH/LHR stimulus also activates the PKA, which phosphorylates phosphodiesterase-5 (PDE5) and hydrolyzes cGMP [64]. Likewise, the LH/LHR stimulus elicits the gene expression of epidermal-like growth factors (EGFs), such as amphiregulin (AREG), epiregulin (EREG), and β-cellulin (BTC). Specifically, LH/LHR decreases the expression of the histone deacetylase 3 (HDAC3), which permits the acetylation of the histone H3 in lysine 14 (H3K14); thus, the SP1 transcription factor can bind to the *Areg* gene promoter, and the synthesis of amphiregulin (AREG) is performed (Figure 4). In humans, the *Areg* gene is in chromosomal region 4q13-4q21, which is flanked by the 5′ region for the *Ereg* gene and the 3′ region for the *Btc* gene, so it is probable that the gene expression of *Ereg* and *Btc* would be similar to that of the *Areg* gene [65]. After the secretion of the EGFs, they act on MCs and CCs through their respective receptors (EGFRs). In MCs, EGFs/EGFRs trigger the ERK1/2 pathway to activate the transcription factors ELK-1 and EGR1, which regulate the expression of tristetraprolin (TTP), which is an mRNA binding protein that binds to the 3′UTR of the *Nppc* mRNA on the ARE motifs, which induce Nppc mRNA degradation [66]. Simultaneously, the retinol binds on the STRA6 in MCs, passing into the cytoplasm to produce retinoic acid (RA), which migrates to the nucleus to bind with its receptor RAR. Then, the RA/RAR complex synergizes with ELK-1 and EGR1 to induce TTP synthesis (Figure 4). Hence, the level of peptide CNP rapidly decreases [67]. On the CCs, the EGFs/EGFRs activate the ERK1/2 pathway to phosphorylate the connexins-37 that close the gap junctions, preventing the cGMPs from going toward the oocyte [68]. Without CNP/NPR2 interactions, cGMP production is decreased. The closing of the gap junctions and the low level of cGMP conduce to PDE3A activation in the oocyte, which degrades the cAMP to AMP, reducing the PKA levels and producing a loss in the phosphorylation status of WEE1B and CDC25. Subsequently, WEE1B becomes inactive and cannot maintain its suppression over CDK1, and CDC25 turns on its activity and dephosphorylates CDK1. Thus, CDK1 is converted into a catalytically active enzyme in conjunction with cyclin B, which results in the meiotic resumption that progresses until metaphase II, when the oocyte reaches its competence and is ready to be fertilized by a sperm [48,68]. Along with the above, the estradiol binds to the membrane G-protein-coupled estrogen receptor (GPER), which activates the ERK1/2 pathway to decrease the protein expression levels of TZP assembly-related genes, which decreases the number of TZPs and subsequently leads to the loss of COC communication [69] (Figure 4). Furthermore, the stimulus of LH/LHR elicits the RAS-RAF-MAPK pathway, which turns on the expression of prostaglandin-endoperoxide synthase 2 (PTGS2), hyaluronan synthase 2 (HAS2), and tumor necrosis factor-alpha induce protein 6 (TNFAIP6), which are indispensable proteins for the initiation of the ovulation process [48].

## 3. Perspectives and Conclusions

It is necessary to improve the quality of embryos produced by in vitro protocols in cattle and humans. The data from the IETS are precise and evidence that over time, there has been an improvement in the use of in vitro embryo protocols, producing billions of embryos. Thus, extensive knowledge of reproduction physiology has been acquired in the last three decades. The use of assisted reproductive technology (ART) has permitted the generation of at least 12 million babies since 1978, when the first baby was born [70]. However, despite this success, the percentage of pregnancies per cycle continues to be low, with a 30% success rate [1,2]. Successful embryo formation depends on the oocyte from which an embryo is formed. So, the quality of oocytes is an essential factor in producing in vitro embryos. Understanding the factors leading to the quality of the oocyte in the ovary is vital if the goal is to improve the percentage of successful pregnancies in women by ART. Nonetheless, obtaining the ovary cellular material is complex, and ethical concerns limit the study of women’s ovarian physiology. In this regard, using oocyte in vitro maturation (IVM) protocols is unconventional. Generally, a woman is subject to a tedious protocol of hyperstimulation, which consists of the administration of high doses of gonadotropins through a series of injections during a previously established period, which results in the stimulation of multiple follicles with the intra-follicular maturation of the oocytes. Posteriorly, a follicular aspiration process is executed to obtain the mature oocytes, which will be fertilized with sperm by a culture protocol or by intra-cytoplasmic sperm injection (ICSI). In cattle, the in vitro maturation of oocytes is a frequently used protocol, and the standardization of this protocol has been established in many laboratories, principally using the ovaries of cows from slaughterhouses.

As mentioned, the physiology process involved in early embryonic development is an event remarkably preserved through mammals of eutherian species. Specifically, women and cows share significant similarities (Table 1). Both have ovarian structures of similar size and the same follicle stages, from primordial to antral. Also, both have two to three follicular waves with a similar number of antral follicles and a dominant follicle is selected, which implies the ovulation of only one oocyte per cycle, so both are monovular. Their oocyte maturation times are similar: 20 to 24 h in cows and 24 to 40 h in women [71]. Also, their cycles of ovarian activity are significantly similar: the estrous cycle of a cow takes about 21 days, while the menstrual cycle in women lasts 28 days. Both have oocytes that are in a state of prolonged meiotic arrest for years, culminating in puberty. Likewise, in both, the FSH controls the recruitment of follicles for a follicular wave, and the dominant follicle has an LH dependence. The time between follicular activation and reaching the preantral follicle stage is approximately 6 months in both species. Finally, both have a gestation period of around 9 months [72]. Thus, the cow is an excellent animal model for studying human ovary physiology because it allows us to apply all the years of experience accumulated using ART without the ethical limitations of humans. Also, it is an easy-to-handle and readily available animal from which large quantities of reproductive tissue can be obtained. Since it has not been possible to create an in vitro system that generates oocytes with enough competence to obtain a blastocyst yield close to 80%, there is an opportunity to improve embryo quality in both humans and cattle [73,74,75]. The oocyte acquires its competence during the folliculogenesis process, interacting with its immediate environment through crosstalk communication between it and the surrounding follicle cells. Understanding how this crosstalk communication impacts the obtention of oocyte quality is crucial to developing in vitro systems that imitate in vivo conditions as closely as possible.

However, despite the bovine model’s benefits in studying human folliculogenesis, there are some limitations. Even if the cows lose their fertility gradually as they age (13–16 years old), eventually becoming infertile, in most cases, the cows are sold to slaughterhouses before reproductive senescence occurs [72,76]. Thus, this model is not feasible for studying menopause in women. Another limitation is that cows develop an estrous cycle, in which the endometrial lining does not undergo periodic detachment during the follicular phase. So, cows do not menstruate to shed the endometrial lining. In contrast to humans, whose reproductive cycle is marked by menstruation, cows’ reproductive cycle is marked by their sexual receptivity [61].
cimb-47-00113-t001_Table 1Table 1Comparative characteristics for reproduction in cows and women.CharacteristicCowsWomenReferencesAge at puberty10–12 months11–12 years[77,78]Length of estrous or menstrual cycle 21 days (estrous)28 days (menstrual)[72]Ovulation after LH peak (hours)119–12 [79,80]Length of gestation period (days)277–290 271–289[78]The average number of offspring11[61]Ovary volume (cm^3^)5–66.6[81,82]Average of oocytes at birth135,000700,000[18]Follicle primordial diameter (μm)<4035–40[14,33]Follicle primary diameter (μm)40–8050–64[14,33]Follicle secondary diameter (μm)81–130115–125[14,33]Follicle tertiary (μm)250–500150–250[14,33]Follicle dominant diameter (mm)8.5–1010–29[79,83]Oocyte metaphase II diameter (μm)115–120103–119[84,85]Number of follicular waves2–3 2–3[61]Number of follicles per wave8–414–14[61]Genome embryonic activation 8–16 cells4–8 cells[86]Blastocyst formation (days)75[86,87]


## Figures and Tables

**Figure 1 cimb-47-00113-f001:**
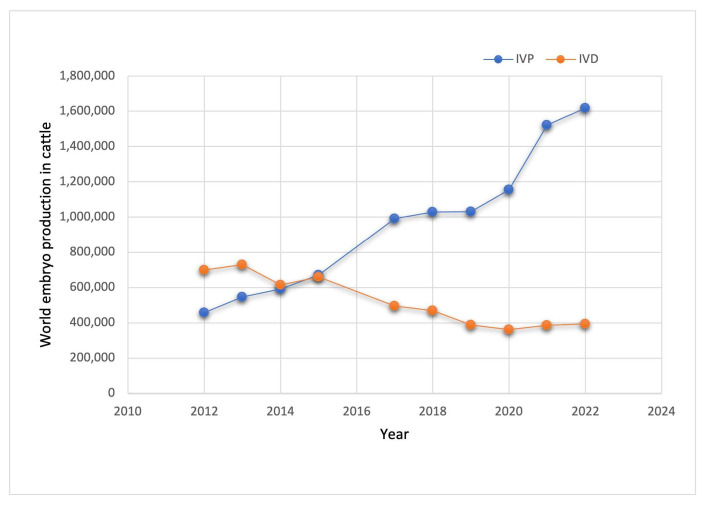
World embryo production in cattle over the last fourteen years. IVP: in vitro production. IVD: in vitro embryo-derived. Data were obtained from IETS, https://www.iets.org, (accessed on 20 August 2024).

**Figure 2 cimb-47-00113-f002:**
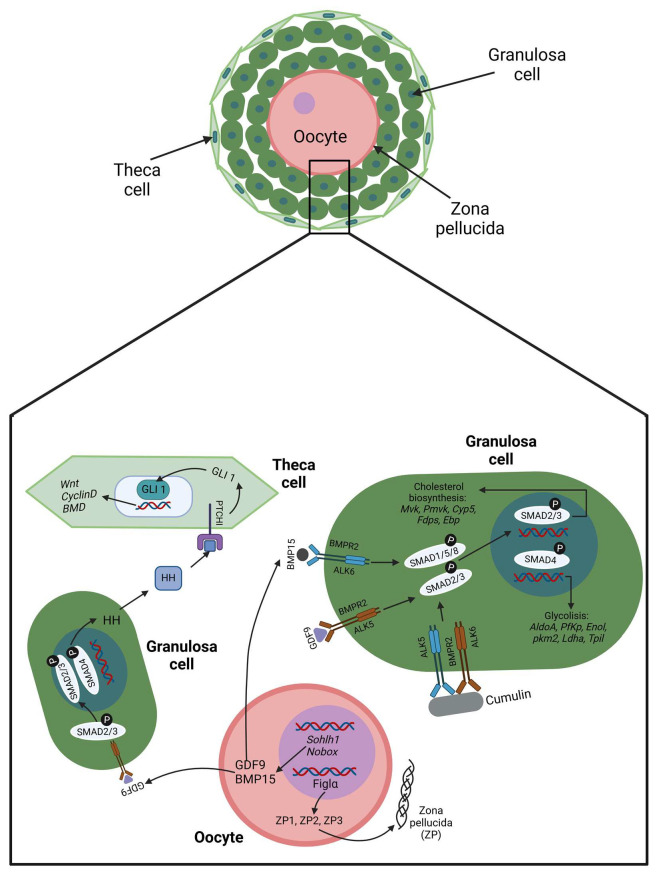
Cellular crosstalk in a primary follicle. The graphical abstract was created using BioRender. Velarde, A (2025); https//Biorender.com/n26b479 (accessed on 2 January 2025).

**Figure 3 cimb-47-00113-f003:**
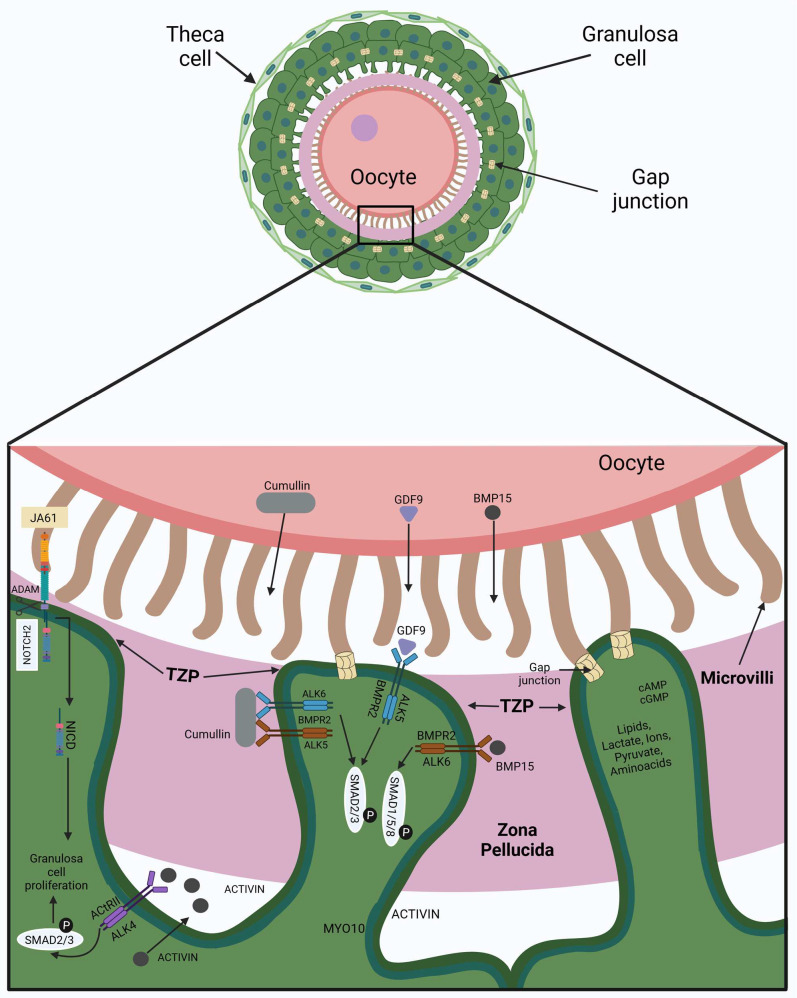
Cellular crosstalk in a secondary follicle. The graphical abstract was created using BioRender. Velarde, A (2025); https//Biorender.com/n26b479 (accessed on 2 January 2025).

**Figure 4 cimb-47-00113-f004:**
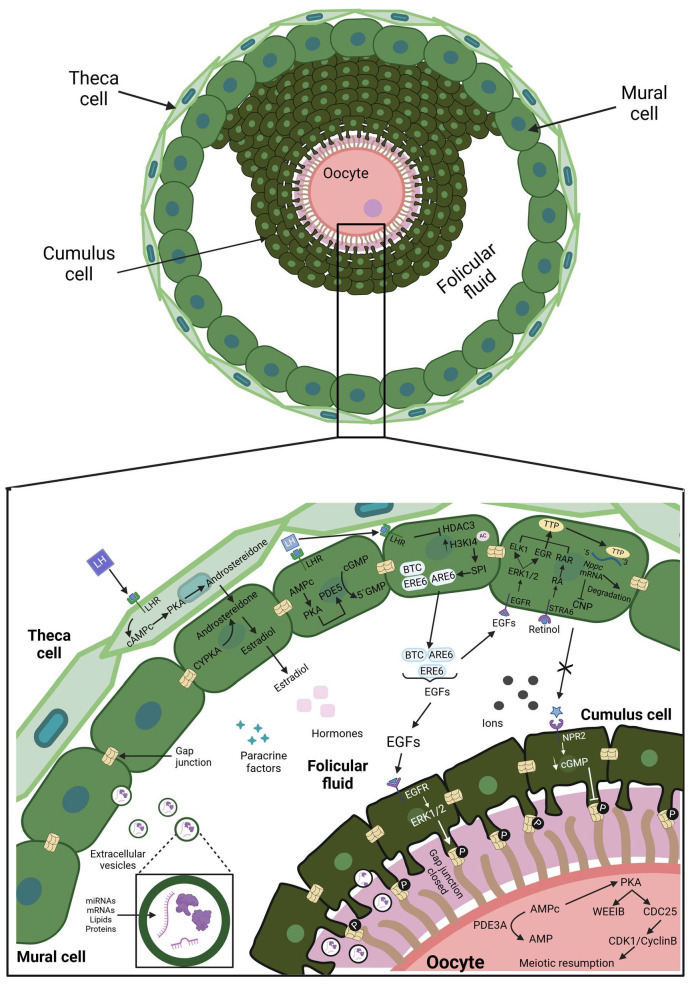
Cellular crosstalk in an antral follicle. The graphical abstract was created using BioRender. Velarde, A (2025); https//Biorender.com/n26b479 (accessed on 2 January 2025).

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
