# Peer review of "Folliculogenesis: A Cellular Crosstalk Mechanism"

_cimb, 2025, doi:10.3390/cimb47020113_

Round 1
Reviewer 1 Report
Comments and Suggestions for Authors
The paper is very interesting, but the main shortcoming is the lack of clear information on which data are from human folliculogenesis and which are from cow folliculogenesis. It is necessary to separate this information very precisely, because the reader may get the wrong impression and draw far-reaching conclusions that may result in attempts at clinical action on the basis of data that are not obvious in human reproduction.
The limitations of the bovine model in translational research should bve considered.
"success of oocyte competence" is unclear and needs to be clarified.
Over-simplification in the description of the influence of hormonal factors on specific stages of folliculogenesis.
Despite the high quality of the graphics, not all are sufficiently described in the text (e.g. there is no reference to diagrams in the relevant sections).
272-273 There is no detailed description of the mechanism of atresia of the remaining vesicles.
95 - Graaff follicle[12]
99 - e primordial follicle[13]
266 - increase of increase?
275 - action[41].
Author Response
- Comments 1: The paper is very interesting, but the main shortcoming is the lack of clear information on which data are from human folliculogenesis and which are from cow folliculogenesis. It is necessary to separate this information very precisely, because the reader may get the wrong impression and draw far-reaching conclusions that may result in attempts at clinical action on the basis of data that are not obvious in human reproduction.
Response 1: We are very agree with this observation. Therefore, we added specific information for folliculogenesis in cows and women throughout the manuscript. The Reviewer can find this new information highlighted in yellow.
- Comments 2: The limitations of the bovine model in translational research should bve considered.
Response 2: We are grateful for this suggestion. In lines 454 – 462, we add the limitations of using bovine as a model for human folliculogenesis. This information is highlighted in yellow:
“However, despite the bovine model's benefits in studying human folliculogenesis, there are some limitations. Even if the cows lose their fertility gradually as they age (13 – 16 years old), eventually becoming infertile, in most cases, the cows are sold to the slaughterhouse before the reproductive senescence occurs [73,77]. Thus, this model is not feasible for studying women's menopause. Another limitation is that cows develop an estrous cycle, in which the endometrial lining does not undergo periodic detaching during the follicular phase. So, cows do not have menstruation to shed the endometrial lining. In contrast to human, where the reproductive cycle is marked by menstruation, in cows, this is manifested by their sexual receptivity [62]”
- Comments 3: "success of oocyte competence" is unclear and needs to be clarified.
Response 3: The sentence was rewritten in lines 15 – 17:
“This review focuses on crosstalk communication during folliculogenesis to deeply understand the events involved in developing oocyte competence necessary to generate an embryo after fertilization.”
- Comments 4: Over-simplification in the description of the influence of hormonal factors on specific stages of folliculogenesis.
Response 4: We expand the information throughout the manuscript.
- Comments 5: Despite the high quality of the graphics, not all are sufficiently described in the text (e.g. there is no reference to diagrams in the relevant sections).
Response 5: Suggestion attended.
- Comments 6: 272-273 There is no detailed description of the mechanism of atresia of the remaining vesicles.
Response 6: A description of atresia mechanis was added in lines 357 – 361:
“The atresia is a mechanism that refers to the degeneration and resorption of follicles by a noninflammatory process before ovulation. The most common form of atresia occurs through apoptosis, although it also can be provoked by the degenerative process of necrosis [54]”
- Comments 7: 95 - Graaff follicle[12]
Response 7: Done
- Comments 8: 99 - e primordial follicle[13]
Response 8: Done
- Comments 9: 266 - increase of increase?
Response 9: Done
- Comments 10: 275 - action[41].
Response 10: Done
Reviewer 2 Report
Comments and Suggestions for Authors
The authors present a comprehensive review of communication during folliculogenesis, emphasizing the processes that contribute to oocyte competence, which is crucial for successful embryo formation following fertilization. This review is quite meaningful, but several issues need further modification.
1.The manuscript's content comes across as somewhat basic and lacks discussion on current research advancements. I recommend that the authors include the latest findings in the introduction or discussion section to enhance the article's timeliness and academic value. Additionally, the authors should discuss the limitations of current research and possible future research directions. For instance, which areas of research are well understood, which remain unclear, and where there are species differences.
2 . Many contents of the manuscript are based on studies with model organisms, yet the authors do not clarify how these studies apply to cattle or humans. I suggest that the authors explicitly state the relevance of mouse studies to cattle and humans and discuss possible species differences.
3. This manuscript implies a comparison between cattle and humans, but it lacks this comparison in the actual content. While it mentions cattle as a model for studying human ovarian physiology, it does not provide specific comparative data and analysis. I recommend that the authors include a discussion of the specific comparisons between cattle and humans in the processes of folliculogenesis and oocyte maturation to enhance the manuscript's relevance and depth.
Author Response
- Comments 1: The authors present a comprehensive review of communication during folliculogenesis, emphasizing the processes that contribute to oocyte competence, which is crucial for successful embryo formation following fertilization. This review is quite meaningful, but several issues need further modification.
Response 1: We appreciate the Revierwer's comment
- Comments 2: 1.The manuscript's content comes across as somewhat basic and lacks discussion on current research advancements. I recommend that the authors include the latest findings in the introduction or discussion section to enhance the article's timeliness and academic value. Additionally, the authors should discuss the limitations of current research and possible future research directions. For instance, which areas of research are well understood, which remain unclear, and where there are species differences.
Response 2: We appreciate this opinion, so we enriched the manuscript and included the latest findings. The Reviewer can read this, which is highlighted in yellow.
In lines 454 – 462, we add the limitations of using bovine as a model for human folliculogenesis. This information is highlighted in yellow.
“However, despite the bovine model's benefits in studying human folliculogenesis, there are some limitations. Even if the cows lose their fertility gradually as they age (13 – 16 years old), eventually becoming infertile, in most cases, the cows are sold to the slaughterhouse before the reproductive senescence occurs [73,77]. Thus, this model is not feasible for studying women's menopause. Another limitation is that cows develop an estrous cycle, in which the endometrial lining does not undergo periodic detaching during the follicular phase. So, cows do not have menstruation to shed the endometrial lining. In contrast to human, where the reproductive cycle is marked by menstruation, in cows, this is manifested by their sexual receptivity [62]”
- Comments 3: 2 . Many contents of the manuscript are based on studies with model organisms, yet the authors do not clarify how these studies apply to cattle or humans. I suggest that the authors explicitly state the relevance of mouse studies to cattle and humans and discuss possible species differences.
Response 3: We are very agree with this observation. Therefore, we added specific information for folliculogenesis in cows and women throughout the manuscript. The Reviewer can find this new information highlighted in yellow.
- Comments 4: 3. This manuscript implies a comparison between cattle and humans, but it lacks this comparison in the actual content. While it mentions cattle as a model for studying human ovarian physiology, it does not provide specific comparative data and analysis. I recommend that the authors include a discussion of the specific comparisons between cattle and humans in the processes of folliculogenesis and oocyte maturation to enhance the manuscript's relevance and depth.
Response 4: We value the Reviewer's observation. We extend the comparative details between cows and women throughout all manuscripts. The Reviewer can read this added and highlighted in yellow.
Reviewer 3 Report
Comments and Suggestions for Authors
This review summarises the role of follicle cells in supporting ovary and oocyte maturation in variable stages of adulthood, and has a specific focus on studies in cow and human. This review is well-rewritten and covers key concept, illustrate key signalling molecule cross-talk between follicle and oocyte, and covers sufficient amount of relevant research. I have not found any significant concerns in the content, I therefore endorse the manuscript for publication.
Author Response
- Comments 1: This review summarises the role of follicle cells in supporting ovary and oocyte maturation in variable stages of adulthood, and has a specific focus on studies in cow and human. This review is well-rewritten and covers key concept, illustrate key signalling molecule cross-talk between follicle and oocyte, and covers sufficient amount of relevant research. I have not found any significant concerns in the content, I therefore endorse the manuscript for publication.
Response 1: We sincerely appreciate the reviewer’s comment.